# Polyphenol Extracts from Sage (*Salvia lavandulifolia* Vahl) By-Products as Natural Antioxidants for Pasteurised Chilled Yoghurt Sauce

**DOI:** 10.3390/antiox12020364

**Published:** 2023-02-03

**Authors:** Cristina Cedeño-Pinos, Antonia María Jiménez-Monreal, María Quílez, Sancho Bañón

**Affiliations:** 1Department of Food Technology and Science and Nutrition, Veterinary Faculty, Regional Campus of International Excellence “Campus Mare Nostrum”, University of Murcia, 30100 Murcia, Spain; 2CIBER: CB12/03/30038 Pathophysiology of Obesity and Nutrition, CIBERobn, Carlos III Health Institute (ISCIII), 28013 Madrid, Spain; 3Research Group on Rainfed Crops for the Rural Development, Murcia Institute of Agri-Food Research and Development (IMIDA), 30150 Murcia, Spain

**Keywords:** aromatic plants, antioxidants, antimicrobials, phenolic acids, flavonoids, clean label

## Abstract

Sage by-product extracts (SE) are a valuable source of phenolic acids and flavonoids for food applications. The objective was to test two SE as antioxidants in pasteurised chilled yoghurt sauces against oxidation. Two SE of different polyphenol total content and profile were selected: SE38 (37.6 mg/g) and SE70 (69.8 mg/g), with salvianic and rosmarinic acid as the main polyphenols, respectively. Four experimental low-fat yoghurt sauces were formulated: untreated; SE70/2 (0.16 g/kg); SE38 (0.3 g/kg); and SE70 (0.3 g/kg). The stability of phenolic acids, microbiological quality (mesophilic bacteria, moulds and yeasts, and *L. monocytogenes*), and oxidative stability (lipids, colour, and pH) were studied in the sauces after pasteurisation at 70 °C for 30 min (day 0) and stored by refrigeration (day 42). Pasteurisation and further chilling ensured the microbiological quality and inhibition of microbial growth could not be evidenced, although SE70 showed some antimicrobial potential. Both SE showed good properties as antioxidants for yoghurt sauces. This finding was based on two results: (i) their main polyphenols, salvianic and rosmarinic acids, resisted to mild pasteurisation and remained quite stable during shelf life; and (ii) SE improved radical scavenging capacity, delayed primary and secondary lipid oxidation, and increased colour stability, contributing to sauce stabilisation. SE38 had a better antioxidant profile than SE70; therefore, the selection criteria for SE should be based on both quantity and type of polyphenols. Due to their stability and antioxidant properties, sage polyphenols can be used as natural antioxidants for clean-label yoghurt sauces.

## 1. Introduction

There is growing interest in using natural ingredients for food with preservative properties and/or health benefits. Among the many natural products with antimicrobial and antioxidant properties, the phenolic extracts obtained from fruits, plants, and other vegetables stand out for their potential applications. In this regard, the distillation industry of essential oils from aromatic medicinal plants (AMP) generates by-products rich in polyphenols that can be exploited as natural preservatives [1,2] or functional ingredients, thus contributing to the circular economy and reducing the environmental impact. Until a few years ago, distillation by-products of some AMP (rosemary, sage, or thyme) were dried and processed in furnaces to obtain energy in factories. Over time, some by-products, such as the distilled rosemary leaf, began to be revalorised as sources of some polyphenols (carnosic acid and carnosol) that are included in the positive lists of food additives [3]. However, others, such as the sage (*Salvia* spp.) by-product, remain practically useless for this purpose, despite possessing good technological and nutritional properties. Sage distilled leaf can retain relevant quantities of bioactive compounds that are not removed with water steam, including phenolic acids, flavonoids, and terpenes [2,4]. These compounds can be recovered with sage by-product extracts (SE) and may present an efficacy similar to that of synthetic antioxidants [5]. For instance, some polyphenols present in sage by-product, such as rosmarinic, salvianic, and salvianolic acids, can act as radical scavengers and metal chelating agents [4,6]. Nevertheless, these antioxidants can participate in more than one reaction (donating an electron and reducing a metal), so that, at low concentrations, they could even behave as prooxidants [7]. Therefore, it is important to test the antioxidant activities by measuring different phases or conditions of oxidation. Many phenolic acids can also exert antimicrobial actions against wide groups of bacteria, moulds, and yeasts, and some, such as benzoic or sorbic acid, are used as food preservatives in the industry. Their antimicrobial properties are related to their ability to alter phospholipidic membranes [8], inhibiting, for instance, in vitro growth of bacteria, such as *Pseudomonas aeruginosa* or *Bacillus cereus* [9]. The use of chia oil (*S. hispanica*) and seeds has been authorised by the European Union (EU 2470, 2017) [10] for human consumption; moreover, the US Food and Drug Administration (FDA) has recognised essential oils, oleoresins (without solvents), and natural extracts (including distillates) of some sage species (*S*. *officinalis*, *S*. *fruticosa*, *S*. *lavandulifolia*, and *S*. *sclarea*) as safe for the food and pharmaceutical industry. Some SE have shown positive effects on Alzheimer’s disease [11], cholesterol reduction [12], and in vitro anticancer agents [13].

SE can be introduced in food either as antioxidants or functional ingredients. A recent study on candies [14] revealed that sage’s (*S. lavandulifolia* Vahl) main phenolic acids (rosmarinic and salvianic acid) are quite stable upon cooking conditions, maintaining their antioxidant properties intact, which provides a good opportunity for developing products enriched with polyphenols. In this study, the antioxidant actions of SE depended on the doses and types of phenolic acids and flavonoids present in the SE. Other trials reported that sage essential oils and extracts contribute to maintaining the oxidative stability of lipids in poultry pâté [15], fresh pork sausages [2], salmon patties [16], goat’s milk-based beverages [17], and salad dressings [5]. Furthermore, they decrease the formation of biogenic amines in sardine fillets [18] and can inhibit the growth of *Salmonella* spp., *Escherichia coli*, and *Listeria monocytogenes* in pork sausage [2] and of *Staphylococcus aureus* in salmon patties [16]. From these studies, it can be assumed that sage polyphenols are stable enough to act as antioxidants and/or antimicrobials during shelf life.

Sauces are ready-to-eat products that are widely consumed around the world and have become important both nutritionally and economically [19]. In particular, commercial yoghurt sauces are often caloric products (up to 406 kcal or 1676 kJ/100 g), due to their high content of lipids (16–42 g/100 g) and simple carbohydrates (6–10 g/100 g [20]. These sauces are prepared with vegetable oil emulsified with milk proteins and stabilised with thickeners, which may also contain salt, spices, flavours, and food additives. Recently, several strategies based on fat replacement by dietary fibres (chicory inulin and apple pectin) and by corn starch [20,21] have been proposed to obtain less-caloric yoghurt sauces. Due to their ingredients, these sauces are prone to microbiological and oxidative deterioration, generating off-flavours that decrease shelf life [7], requiring thermal and/or chilling treatments. Microbial load and lipid oxidation are often controlled in sauces with preservative additives (sulphite, butylated hydroxy toluene, or butylated hydroxy anisole) that are currently being questioned, owing to their possible detrimental effects on human health [22]. Different natural extracts from rosemary [20], sage and oregano [5], citrus [23], olive leaf [24], grape pomace [25], rosemary, and thymus [26] have been tested as natural preservatives in clean label food products. Some of these trials have revealed that plant extracts may provide unfavourable sensory traits (herbal notes, bitterness, astringency, or darkness) to food. In a previous trial, a yoghurt sauce with inulin was formulated with micro quantities of rosemary extracts (at 0.3 g/kg) that enhanced the antioxidant properties, although microbiological and oxidative stability were not studied [20]. Aqueous SE have a slight camomile flavour, are less bitter, and less astringent than rosemary extracts rich in rosmarinic acid [14] and could be used at the above-mentioned concentration with a lesser risk of transferring herbal off-flavours to yoghurt sauces. The research hypothesis was that introducing micro quantities of SE can enhance the antioxidant properties of this type of sauces.

The objective of this work was to assess two selected SE of different polyphenol total content and profile to stabilise a pasteurised chilled yoghurt sauce against oxidation.

## 2. Materials and Methods

### 2.1. Experimental Design

A randomised design was performed with two different SE: (i) SE38 (37.64 mg polyphenols/g) and (ii) SE70 (69.82 mg polyphenols/g), with salvianic and rosmarinic acids as the main polyphenols, respectively. Four experimental low-fat yoghurt sauces with inulin were compared: (i) untreated; (ii) SE70/2 (0.16 g/kg); (iii) SE38 (0.3 g/kg); and (iv) SE70 (0.3 g/kg). This design enabled comparison of: (i) same concentrations (0.3 g/kg) of different extracts (SE38 vs. SE70); (ii) same concentrations of total polyphenols (11 mg/kg) from different extracts (SE38 vs. SE70/2); and (iii) two different concentrations (0.3 vs. 0.16 g/kg) of the same extract (SE70/2 vs. SE70). Stability of sage polyphenols, microbiological quality, antioxidant capacity, and oxidative stability were studied in the sauces after pasteurisation (day 0) and refrigerated storage (day 42). Descriptive statistics were used to study sauce levels of salvianic and rosmarinic acids. One-way ANOVA was used to determine the effects of SE addition on total viable counts and ΔE* values. Two-way ANOVA was used to determine the effects of SE addition and chill storage on the dependent variables. Sample size per each treatment level was *n* = 9 (3 sauce jars × 3 manufacturing batches). The Tukey’s rank test was used (*p* < 0.05). Data were analysed with the Statistix 8.0 software for Windows (Analytical Software, Tallahassee, FL, USA).

### 2.2. Obtention and Typification of Sage By-Product Extracts

Two experiment extracts were obtained from the respective sage (*Salvia lavandulifolia* Vahl) ecotypes, selected according to their yield in polyphenols in the laboratory of the Rain Feed Crops for the Rural Development Department, which forms part of the Institute for Agrifood and Environmental Research and Development located in Murcia, Spain [14]. Cloned sage plants were cultivated to obtain plant material. Essential oil was previously removed from the sage leaves by distillation with water steam [27]. SE were obtained by extraction with water of oil-free distilled leaf. Oil-free dry leaves were ground to 2 mm to obtain the powder by-product. Sage powder was mixed with water at 1:10 (*w*:*v*) ratio, kept at 30 °C for 90 min in a water bath with constant stirring, and centrifuged at 4560× *g* and 5 °C for 10 min in a Digecen 21 R centrifuge (Orto Alresa, Madrid, Spain). The supernatant was filtered (Whatman No. 4), lyophilised (Lyobeta 15, Telstar) at 100 mbar and −80 °C for 24 h, packed into a dark steel container with nitrogen, and kept at −80 °C and in the dark until further use. SE powders had a green-brown colour (see graphical abstract). The values of CIELab colour and pH of extracts were checked. SE38: L* = 58.71; a* = 5.60; b* = 20.45; and pH = 6.8; and SE70: L* = 52.18; a* = 4.97; b* = 17.53; and pH = 6.4.

Polyphenol composition of both SE was typified according to Jordán et al. (2013) [27]. Methanolic solutions of SE samples were analysed using an HPLC-1200 Series (Agilent, Waldbronn, Germany) equipped with a G1311A binary pump and a G1315A UV/Vis photodiode array detector. Identification was performed by comparing retention times with the respective spectra of different commercial standards. Quantification was carried out using linear regression models based on standard dilution techniques. For more detailed information concerning chromatographic conditions, chemical standards and identification and quantification methods, see Jordán et al. (2013) [27] and Martínez-Tomé et al. (2022) [20]. Thirteen polyphenols were quantified in SE (mg/g extract), including six phenolic acids and seven flavonoids (see profiles in Figure 1) [20]. The ratio between phenolic acids and flavonoids was more equilibrated for SE38 (2.0) than for SE70 (3.1). The AC was checked in SE samples at the concentrations used in sauces, as described in Section 2.5.

### 2.3. Elaboration of Yoghurt Sauce

Sauce ingredients were purchased from a local supermarket in Murcia (Spain). The yoghurt sauces were manufactured with slight modifications at the Food Technology Pilot Plant at the University of Murcia (Spain), as described by Martínez-Tomé et al. (2022) [20]. The ingredients and their proportions are listed in Table 1. Sauce ingredients were mixed and homogenised (2 min; 25 °C) using a cooking robot (Taurus Mycook 1.6, Lérida, Spain). The homogenate was mixed with water (untreated) or SE water solution of (SE-enriched) for 1 min to obtain raw sauce. Raw sauce was packed in 33 mL glass jars (41 mm height × 40–43 mm diameter) with a metallic lid (see graphical abstract), pasteurised (70 °C; 30 min; reaching 70 °C at 10.2 min in the centre), and cooled (5 °C; 30 min) in the respective water baths with constant stirring.

Pasteurised sauce was kept for 42 days in a 103899 Difri display cabinet (Valencia, Spain) (4 °C; 600 lx continuous white lighting). All samples (day 0 and 42) were kept at −80 °C until further analysis. The proximate composition (g/100 g ± SEM; *n* = 9) of yoghurt sauce was: 69.38 ± 0.11 (moisture), 15.84 ± 0.18 (carbohydrates), 11.25 ± 0.06 (lipids), 2.47 ± 0.02 (proteins), and 1.06 ± 0.01 (ash), providing 174.49 kcal or 730.06 kJ per each 100 g sauce. Analytical procedures (ISO Norms and AOAC methods) can be consulted in Martínez-Tomé et al. (2022) [20].

### 2.4. Microbiological Analyses

Total viable counts (Log CFU/g) were determined in plate count agar at 30 °C, according to the ISO 4833-1:2013 [28]. Total counts of moulds and yeasts (log CFU/g) were determined by an in-house method based on ISO 21527-2:2008 [29] (plate count: DG-18; agar at 25 °C). *Listeria monocytogenes* (absence in 25 g product) was determined by the method: MET-Mi-*L. monocytogenes*—Al (detection: ALOA Agar) ISO 11290-1:2017 [30].

### 2.5. Antioxidant Capacity and Polyphenols Analyses

Methanolic extracts of sauce samples were prepared for AC tests and the quantification of phenolic acids. A total of 3 g sauce was mixed with methanol to 10 mL using a calibration flask. The diluted sample was constantly shaken in the dark (25 °C; 10 min) and centrifuged (2580× *g*; 25 °C; 10 min) in a D2010 centrifuge (Kubota, Tokyo, Japan). The supernatant was filtered with a Whatman No. 1 paper and stored at −80 °C until further use. Rosmarinic and salvianic acids were quantified (mg/kg sauce) by HPLC-DAD, as described in Section 2.2. Initial quantities of both phenolic acids used in sauces were calculated from SE quantification. Total phenolic content (TPC) was determined with the Folin-Ciolcateu method [31]. Sample absorbance was measured at 760 nm using a UV-VIS spectrophotometer (Genesis 180, Madison, WI, USA). The calibration line used for quantification ranged from 1–6 µg gallic acid GA/mL (R^2^ = 0.9999). Results were expressed as mg GAE/100 g sauce. The 2,20-azino-bis-(3-ethylbenzothiazoline-6-sulphonic acid) (ABTS) decolourisation assay was carried out by spectrophotometry (734 nm) [32]. Quantification was made with a calibration line (R^2^ = 0.9997) prepared with ±-6-hydroxy-2,5,7,8-tetramethylchromane-2-carboxylic acid (Trolox) at 0.1–5 µg Trolox/mL. Results were expressed as mg Trolox equivalents TE/100 g sauce. The 2,2-diphenyl-1-picrylhydrazyl radical (DPPH•) assay was performed by spectrophotometry (517 nm) [33]. The calibration line for quantification ranged from 0.1–5 µg Trolox/mL (R^2^ = 0.9991). Results were expressed as mg TE/100 g sauce. The ferric reducing antioxidant power (FRAP) was determined by spectrophotometry (593 nm) [34]. The calibration line for quantification ranged from 100–700 µmol Fe^2+^ (FeSO_4_·7H_2_O). Results were expressed as µmol Fe^2+^ equivalents/100 g sauce. The detailed methodology followed for different AC tests is available at Cedeño-Pinos et al. (2023) [14].

### 2.6. Lipid Oxidation Indexes

The lipid fraction of sauce was extracted by the Folch method with some modifications [35]. A total of 20 g sauce was mixed with chloroform: methanol (1:1 *v/v*), contained 0.003% (*w/v*) butylated hydroxytoluene as antioxidant. Next, 1 M KCl was added, and the organic phase was separated and dried under vacuum (30 °C and 337–470 mm Hg pressure) in a Hei-vap rotary evaporator (Heidolph, Schwabach, Germany). Lipid content was gravimetrically determined and stored at −80 °C for further analysis. The analytical procedures used are available in detail in the work of Cedeño-Pinos et al. (2022) [36]. Conjugated dienes (CD) and trienes (CT) were determined according to Pegg (2001) [37]. The absorbance of sample extract in 2,2,4-trimethylpentane (isooctane) (0.01 g/mL) was measured at 233 nm (CD) or 268 nm (CT) using a UV/Vis spectrophotometer (Spectronic Unicam, New York, NY, USA). Results were expressed as µmol/g sauce. Peroxide values were also determined (AOCS, Cd 8-53) [38]. A total of 1 g lipid fraction was homogenised (1 min) with 10 mL chloroform, 15 mL glacial acetic acid, and 1 mL saturated aqueous potassium iodine solution. The mixture was kept in the dark for 5 min, and 75 mL distilled water was then added and titrated with 0.002 N sodium thiosulphate solution in an automatic titrator equipped with a 0160451100 Pt WOC combination electrode (Metrohm Hispania, Madrid, Spain). Results were expressed as meq O_2_/kg lipids. Thiobarbituric acid reactive substances (TBARS) were determined according to Botsoglou et al. (1994) [39]. A total of 2 g sauce was homogenised with 8 mL trichloroacetic acid water solution (5% *w*:*v*) and 5 mL butylated hydroxytoluene (in hexane 0.8% *w*:*v*) and then centrifuged (1975× *g*; 25 °C; 10 min). Sample reaction was conducted with 2.5 mL supernatant and 1.5 mL thiobarbituric acid (TBA) solution (0.8% *w*:*v*) (70 °C; 30 min). Absorbance was measured at 532 nm. A calibration line (Y = 1299.8x + 0.0033; R^2^ = 0.999) of malonaldehyde (MDA) prepared with 1.1-3.3-tetraethoxypropane (0.1–10 μM) was used for quantification. Results were expressed as mg MDA/kg sauce.

### 2.7. CIELab Colour and pH Measurements

Colour was measured by reflectance on sauce surface using a Chroma Meter II CR-200/08 (Minolta Ltd., Milton Keynes, UK) with a D65 illuminant, 2° observer angle and 50 mm aperture size. Results were expressed in CIE units: lightness (L*), redness (a*), yellowness (b*), and ΔE* [(L*_day 0_ − L*_day 42_)^2^ + (a*_day 0_ − a*_day 42_)^2^ + (b*_day 0_ − b*_day 42_)^2^]^1/2^. The pH was determined with a MicropH 2001 pH meter (Crison, Barcelona, Spain) using a Cat. 52-22 combined electrode (Ingold Electrodes, Wilmington, DE, USA).

## 3. Results

### 3.1. Antioxidant Capacity of Sage Extracts

The AC was checked in SE samples at concentrations used in yoghurt sauces (Table 2). The highest AC was determined in SE38, followed by SE70 and SE70/2, with some exceptions. TPC (mg GAE/100 g extract) was higher in SE38 and SE70 (around 14) than in SE70/2 (12.5); DPPH value (TE/100 g extract) was also higher in SE38 and SE70 (6–7) than in SE70/2 (4.6). The highest ABTS value (TE/100 g extract) clearly corresponded to SE38 (12.6), followed by SE70 (8.7) and SE70/2 (7.3), while the highest FRAP values (µmol Fe^2+^/100 g extract) again corresponded to SE38 (78.3), followed by SE70 (62.3) and, by far, of SE70/2 (38.2). Overall, the percentage of AC decreased by 2–42% (SE70) and by 13–51% (SE70/2), with respect to the 100% (SE38). SE38 showed better ABTS and FRAP scavenging activities than SE70, when analysed at the same concentrations of extract (0.3 g/kg) or of polyphenols (11 mg/kg). Reducing SE70 concentration from 0.3 to 0.16 g/kg decreased FRAP scavenging activity by one third, although with slightly decreased TPC, DPPH, and ABTS values.

### 3.2. Microbial Quality

The counts of mesophilic bacteria, mould and yeast, and *L. monocytogenes* in the yoghurt sauces are shown in Table 3. Total viable counts were <1 log CFU/g in all sauces at day 0, and they were slightly lower in the SE70 sauce (1.3 Log CFU/g) than in the rest (1.7–1.9 Log CFU/g) at day 42. Mould and yeast counted <1 log CFU/g in all sauces at days 0 and 42. Likewise, *L. monocytogenes* were absent in 25 g sample in all sauces at days 0 and 42. Therefore, pasteurisation (70 °C; 30 min) and chilling (4 °C) widely ensured the microbial quality of yoghurt sauces. Some inhibition of psychrophilic bacteria in SE70 sauce was observed during storage.

### 3.3. Stability of Salvianic and Rosmarinic Acids

Sauce concentrations of salvianic and rosmarinic acids are described in Figure 2. The sauces with SE presented similar levels of both phenolic acids after pasteurisation to those formulated for raw sauce (in parentheses) and then partially degraded with storage. Sauce concentrations (mg/kg) of salvianic acid decreased with storage from 1.2 to 0.01 (1.2) (SE70/2), from 3.2 to 1.27 (3.3) (SE38), and from 2.1 to 0.02 (2.3) (SE70) (Figure 2A). Sauce concentrations (mg/kg) of rosmarinic acid also decreased with storage from 5.6 to 4.0 (6.2) (SE70/2), from 1.8 to 1.6 (1.9) (SE38), and from 10.8 to 8.5 (11.5) (SE70) (Figure 2B). Therefore, both phenolic acids slightly degraded with pasteurisation; salvianic acid virtually disappeared in sauces that were poorer in this acid (SE70/2 and SE70), while rosmarinic acid was quite stable during storage. The accumulated degraded quantities (g/kg) of both phenolic acids in the pasteurised chilled sauces were 2.8 (SE70/2), 2.1 (SE38), and 4.3 (SE70). These results signalled that both phenolic acids contribute toward stabilising yoghurt sauce against oxidation during pasteurisation and further chill storage.

### 3.4. Antioxidant Capacity

The AC of sauces is shown in Table 4. SE38 and SE70 sauces had the highest AC values, followed by SE70/2 and untreated sauces. TPC was not affected by storage—the ABTS and FRAP values decreased with storage, while the DPPH values only decreased with storage in the untreated sauce. There was interaction between SE addition and storage for all AC values, except TPC. TPC (mg GAE/100 g) were higher in the SE38 and SE70 (over 41) than in the untreated sauce (<39) at days 0 and 72. SE70/2 presented intermediate TPC (over 40). Similarly, the DPPH values (mg TE/100 g) were higher in sauces with SE (10.3–12.2) than in untreated sauce (5.0–7.4); the highest DPPH values corresponded to the SE38 (day 42) and S70 sauces (day 0). The ABTS values (mg TE/100 g) were clearly higher in the SE38 sauce at days 0 and 42 (12.6 and 9.6, respectively), followed by the SE70 and SE70/2 sauces (11.9–12.6 and 9.2–9.6) and the untreated sauce (7.7 and 5.5). In contrast, FRAP value (µmol Fe^2+^/100 g) were higher in the SE70 (139.7) than in the SE38 (128.23) sauce at day 0, followed by the SE70/2 (114.9) and the untreated (86.6) sauces, while SE70 and SE 38 presented similar FRAP values (107.9 and 114.4) at day 42, followed again by the SE70/2 (94.4) and untreated (67.7) sauces. SE addition did not improve the antioxidant response to the Folin–Ciolcateu reaction, but clearly enhanced the DPPH, ABTS, and FRAP scavenging activities, as seen for extracts. The ABTS and FRAP values allowed for a better discrimination of SE treatments. SE38 and SE70 (0.3 g/kg) similarly enhanced sauce AC, and the SE38 profile was more efficient than the SE70 profile in enhancing sauce AC, while reducing the SE70 concentration (from 0.3 to 0.16 g/kg) resulted in a slight detriment to the sauce AC.

### 3.5. Oxidative Stability

The lipid oxidation values, CIELab colour, and pH of sauces are shown in Table 5. The SE addition protected sauce against oxidation, which increased with storage. The interaction was seen between both treatments for all variables, except for L* and a*. At day 0, sauces with SE had lower levels of primary (CD, CT, and PV), but not of secondary (TBARS), lipid oxidation values than the untreated sauce; the SE70 and SE70/2 sauces had lower CD and CT levels and more similar PV than the SE38 sauce. At day 42, the sauces with SE had lower CD, PV, and TBARS levels than the untreated sauce; the three sauces with SE reached similar concentrations of CD, CT, PV, and TBARS. The lowest levels of CD + CT (µmol/g) at day 0 and 42 corresponded to the SE70/2 and SE70 sauces (around 1.2), followed by the SE38 (2.1 and 1.0, respectively) and untreated sauces (2.5 and 1.5). These effects were even clearer for PV. The PV (meq O_2_/kg) at days 0 and 42 were lower in the sauces with SE (over 1.2 and 7.5) than in the untreated sauce (1.5 and 14.1). The TBARS values (mg MDA/kg) were similar for all sauces at day 0 (over 0.45) and were slightly lower in those with SE (over 0.55) than in the untreated sauces (0.67) at day 42. Therefore, the three formulations with SE inhibited the formation of hydroperoxides and secondary oxidised lipids in a similar way.

Changes in the colours of the pasteurised sauces were slight but relevant. Sauces with SE had lower L* and higher a* and b* values than the untreated sauce. Sauces turned slightly less bright and somewhat more yellow when adding SE. Chill storage led to some additional discolouration. As shown in Figure 3, changes in CIELab colour (ΔE* values) were less pronounced in SE70 sauces (0.52), followed by SE38 sauces (0.70), SE70/2 (0.88), and untreated (0.93). The pH values of sauces were over 4.0. Unlike those seen for other oxidation parameters, the pH value was similar in sauces formulated with and without SE at day 0 (3.9–4.0) and 42 (over 4.1), except for the SE70, whose pH slightly increased with storage. SE addition contributed toward stabilising the yoghurt sauce colour and had no relevant effects on sauce pH.

## 4. Discussion

Differences in the polyphenol composition of the AMP extracts obtained upon similar extraction conditions are often due to plant material. It was reported that the polyphenol content of sage by-products *(S. lavandulifolia* Vahl and *S. officinalis*) can vary, depending on geographical location and phenological stage of processed plants [4,40]. SE38 was poorer in polyphenols, but contained more salvianic acid (3-(3,4-dihydroxyphenyl) lactic acid), a monomer of caffeic acid whose scavenging activity against free radicals (HO hydroxyl and O^2−^ superoxide anion) is even superior to that of vitamin C [41]. Salvianic acid is the main antioxidant of dansenshu (*Salvia miltiorrhiza*) aqueous extracts, a Chinese medicinal herb [6], and is also present in the aqueous extracts of sage [4], lemon balm [42], and rosemary [43]. SE70 was richer in total polyphenols, particularly in rosmarinic acid, a derivative of caffeic acid and synthesised by ester linkage with 3,4-dihydroxyphenyl lactic acid [6]. This acid is considered the most abundant phenolic acid in most SE [44], being responsible for different biological activities, including antioxidant, antimicrobial, antiviral, enzyme inhibitory, and anticancer [45]. Both SE38 and SE70 showed AC (assessed as TPC; DPPH, ABTS, and FRAP) at the concentrations used in the sauce (0.16 or 0.3 g/kg). Depending on the assay used, SE38 showed better or similar AC than SE70, likely due to the fact that the ratio between phenolic acids and flavonoids was more balanced in SE38 (2.1) than in SE70 (3.0). Synergies between phenolic acids (caffeic and rosmarinic acids) and flavonoids can enhance the AC of polyphenol mixes [46]. SE70 concentration had a moderate effect on the resulting AC. This could be due to several reasons. Antioxidant reactions may be affected to a greater or lesser extent by the concentration of key analytes. Moreover, gallic acid or Trolox standards may not have the same antioxidant response than complex mixes of natural antioxidants in the AC tests, which might explain why the AC values moderately increased when the SE70 concentration was doubled. SE70 samples at 0.16–0.3 g/kg contained 6.2–11.5 and 1.2–2.3 mg/kg of rosmarinic and salvianic acids, respectively. These concentrations were low, compared to those studied in pure compound solutions, where both phenolic acids at 10–320 mg/L scavenged the DPPH radical in a concentration-dependent manner [47]. The obtained results agree with a previous study, where both SE38 showed better AC than SE70 at concentrations of 0.25, 0.50, and 0.75 g/kg [14]. In other studies, several SE also showed good antioxidant properties when tested with different assays (DPPH, ABTS, FRAP, and others) [2,4]. Anti-radical in vitro ability of sage phenolic acids has been compared with that of other well-known antioxidants. For instance, rosmarinic acid and their primary metabolites (caffeic and salvianic acids) show similar DPPH radical-scavenging activity than quercetin [47], while salvianic acid and salvianolic acid B exhibited higher scavenging activities against OH, O_2_, DPPH, and ABTS radicals than vitamin C [6].

Stability of rosmarinic and other phenolic acids was not previously studied in the yoghurt sauce prepared with rosemary extracts [20]. As seen, salvianic and rosmarinic acids practically resisted without degrading to the pasteurisation treatment conducted in sauce (70 °C; 30 min). This is in agreement with results obtained in SE-enriched jelly candies, where levels of both phenolic acids were hardly affected by cooking (80 °C; 10 min) [43]. Rosmarinic acid also showed good thermal stability in baked biscuits (190 °C; 10 min) [48] and baked wheat bread (210 °C; 23 min) [49]. Baked products reach higher temperatures while dehydrating. If rosmarinic acid can resist baking conditions, it is likely that this acid can also resist mild pasteurisation conditions. Therefore, thermal stability does not seem an impediment to using these phenolic acids in heated foods, as occurs with thermolabile antioxidants, such as vitamin C. Both phenolic acids partially degraded in the pasteurised sauce during further storage, perhaps due to oxidation reactions favoured by occluded oxygen and lighting. Forty-two days is a considerable shelf lifetime for a pasteurised yoghurt sauce and chilling likely contributes to keeping the sage antioxidants. The degradation levels of both phenolic acids were consistent with the SE treatment used in sauces. Salvianic acid virtually disappeared in the sauces with SE70 (at 0.16 or 0.3 g/kg), while most of the rosmarinic acid remained undegraded. In contrast, the sauce with SE38 retained relevant levels of salvianic and rosmarinic acid. Due to their chemical structure, phenolic acids can act as secondary antioxidants able to work synergistically by regenerating primary antioxidants, protecting other compounds against oxidation [50]. The bioactivity of phenolic antioxidants depends on configuration of molecules, as well as on the number and position of OH groups [51,52]. The dissociation energy of the OH bond is also decisive, since the lower it is, the more stable the antioxidant radical and the more efficiently it will donate hydrogens [53]. The radical scavenging capabilities of phenolic acids enhance with increasing radical scavenging ring pendant groups [8]. Rosmarinic and salvianic acids should present similar antioxidant actions at the concentration ranges used in yoghurt sauce [47], in such a way that the salvianic acid might be somewhat less stable than the rosmarinic acid in the sauce, perhaps due to other factors influencing the oxidant-antioxidant balance among this type of polyphenols, such as initial concentrations and the existing ratio between both phenolic acids and the action of other sage antioxidants.

Both untreated sauces and those formulated with SE met European Commission microbiological criteria for foodstuffs [54]. The absence (in 25 g) of a psychrophilic pathogen such as *L*. *monocytogenes* and the low counts recorded for mesophiles (<6 Log CFU/g), moulds, and yeasts evidenced that the microbial quality of sauce was largely ensured by the pasteurisation treatment. The pasteurisation conditions were previously checked, and low microbiological counts were expected. Refrigeration maintained these microbiological counts at a low level. In addition, other factors, such as the moderate acidity of sauce (pH = 4) or the possible presence of antimicrobials in honey and mustard, may have contributed to inhibiting microbial growth [55,56]. Due to pasteurisation, the antimicrobial effects of SE on sauce could not be well-elucidated, although some incipient inhibition of mesophilic bacteria was observed in the sauce with 0.3 g SE70/kg at day 42 because the incomplete inhibition of mesophilic bacteria was reached, and this extract was particularly rich in rosmarinic and other phenolic acids. The antimicrobial properties of phenolic acids are related to the presence of ring hydroxyl and/or methoxy groups [8]. Hydroxyl groups of polyphenols can interact with the membrane protein of bacteria via hydrogen bonds and cause changes in membrane permeability and cell destruction [25]. Kostić et al. (2015) [9] reported that the SE (*S. verbenaca*) rich in rosmarinic acid inhibited in vitro growth of *P. areuginosa* and *B. cereus*. An extract (lemon balm) rich in rosmarinic acid showed better antibacterial in vitro effects than streptomycin and ampicillin on several Gram positive and negative bacteria when tested as ingredients for cupcakes [57]. The antimicrobial role of SE would be less relevant in sauces where pasteurisation prevents microbiological risk. However, using SE may facilitate the application of mild thermal treatments or the production of sauces with extended shelf lifetime.

The antioxidant potential of SE was evidenced when determining the AC in yoghurt sauces. This sauce is a complex matrix whose ingredients (yoghurt, honey, mustard, and inulin) contribute to its AC [20,55,56], together with SE. Pasteurisation led to some needed loss of sauce AC, while storage resulted in an additional detriment of AC because oxidation continued in the pasteurised chilled sauce. SE effects on sauce AC could be stabilised using different chemical assays. The Folin–Ciolcateu reagent can positively react with non-phenolic compounds, such as reducing sugars and certain easily oxidisable amino acids [58,59], overestimating the TPC. This may explain why there were no clear differences in the TPC among sauces with SE with different AC, as assessed by other tests. The FRAP, ABTS, and DPPH values, in this order of efficiency, allowed for the discrimination of sauce AC, as seen for extracts. DPPH, ABTS, and FRAP are widely contrasted AC tests used in plant and food samples containing phenolic antioxidants. Sauce AC was enhanced by SE, thus evidencing that those sage antioxidants remained active in the pasteurised chilled sauce, protecting it against oxidation. Reducing SE70 concentration by half only led to a moderate decrease in sauce AC, owing to the probable antioxidant activities of other ingredients. The SE38 profile was more efficient for enhancing sauce AC, reproducing the results obtained in extract samples.

The efficiency of SE as food antioxidants is another important aspect. Yoghurt sauce is an oil-in-water complex emulsion containing oil, pigments, and other components that can oxidise during processing and shelf life, resulting in changes of flavour and colour, among others. Lipid oxidation is stimulated by factors such as lipid unsaturation, oxidising enzymes, radiation, lipid-air interface, occluded oxygen, or the presence of catalysts as metal ions [37,50]. As seen, SE inhibited the formation of primary oxidised lipids (PV, CD, and CT) in the freshly pasteurised sauce. Similarly, Abdalla and Roozen (2001) [5] reported that liposome-encapsulated SE delayed the formation of conjugated dienes and the hexanal production in salad dressings (12 weeks; 20 °C), whose result was comparable to that of BHT liposomes. Lipid oxidation is known to occur through a free radical chain reaction mechanism (lipid radicals, R•; alkoxyls, RO•; peroxyls, ROO•; and hydroxyls, •OH), which is triggered in three stages: initiation, propagation, and termination [50]. Therefore, sauces with SE (with lower initial oxidation rate) are expected to develop lower levels of secondary oxidised lipids during storage. The CD and CT decreased during storage, possibly due to their participation in oxidative reactions, leading to their decomposition or to generation of secondary compounds [37,50]. In contrast, a generalised increase was observed in sauce PV after storage. SE strongly inhibited the formation of hydroperoxides and fulfilled their antioxidant role. Pegg (2001) [37] suggested that, following the induction period, the rate of peroxide formation is exceeded by its rate of decomposition, which would explain the high PV of sauces. This trend is usually reversed at later stages, as during the course of oxidation, PV reach a maximum and then decrease, which would reveal early or advanced oxidation. Phenolic hydroxyl groups, such as those present in sage polyphenols, could reduce hydroperoxide formation by donating hydrogen atoms by scavenging free radicals, thus interfering with the initiation or propagation of lipid oxidation [25,60]. Although some compounds from primary oxidation (such as hydroperoxides) are odourless and colourless, they can degrade into a complex array of secondary products, including aldehydes, ketones, and alcohols, among others (some odorous), generating a negative impact on the flavour [50]. These compounds (assessed as MDA) were hardly detected in the freshly pasteurised sauce, although they did appear after further storage. SE addition inhibited the formation of secondary oxidised lipids, which is coherent with the AC determined in sauces. The formation of secondary compounds is probably related to the oxidation reactions involving the unsaturated lipids contained in sunflower oil and other sauce ingredients, along with other sauce compounds (ketones, amino acids, oxidized proteins, carbohydrate esters, sugars) able to react with TBA [61]. Secondary lipid oxidation levels were incipient, but well below those reported as producing rancidity in meat (>1 mg MDA/kg) [62], being controlled by SE addition. Zhang et al. (2013) [63] also observed a slowdown in TBARS production in sausages fortified with SE (0.1 and 0.15%) after chill storage (21 days; 4 °C). Similarly, Bianchin et al. (2020) [15] found TBARS values of 2.69 and 3.96 mg MDA/kg, respectively, in chicken pâté with and without ethanolic SE after storage (28 days; 4 °C). Lipid oxidation in emulsified sauces is initiated at the oil–water interface, which facilitates the interaction with water-soluble prooxidants and lipids [64]. Specifically, rosmarinic acid can inhibit lipid peroxidation in a concentration-dependent manner (0.25–60 µm) in phospholipidic membranes [65]. The addition of polyphenol, such as those present in SE, with metal chelating activity can physically remove metals from the lipid core and/or the droplet interface, thus preventing degradation of lipid hydroperoxides and, thus, delaying the production of secondary oxidised lipids [7,50]. However, the efficacy of such antioxidants is often conditioned by various factors, such as the concentration, temperature, type of oxidisable substrates, and physical state of the system, as well as by the presence of prooxidants and synergists [50].

Changes in colour and pH are also indicative of sauce oxidation. Yoghurt sauces were white (L* > 85), due to milk emulsion, and presented a dim yellow tonality (low positive values of a* and b*). Sauce whiteness is mainly due to yoghurt and inulin, while sunflower oil, mustard, and SE provide yellow-green tones typical of plant pigments. Sage contains chlorophyll, carotenoid, and polyphenol pigments [66]. In addition, heating can produce non-enzymatic browning reactions and/or transformations of phenolic compounds towards darker compounds [63]. As seen, microquantities of SE were sufficient to produce some changes of reflectance in freshly pasteurised sauce (lower lightness and higher redness and yellowness). These chromatic changes were less evident in the sauce with less quantity of SE70, confirming that sauce colour was affected by sage pigments. Similar changes in CIELab colour were found in a yoghurt sauce with rosemary extract [20], which likely contain the same type of pigments as SE. Sauce darkening suggested that some pigments continued oxidising during storage. White foods containing small quantities of yellow-green plant pigments tend to darken or brown when oxidised, which may affect CIELab values. As seen, sauce CIELab colour remained stable for 42 days with few chromatic changes. The ΔE* values determined in the sauces were modest but indicate that SE can stabilise the colour of this product. Zhang et al. (2013) [63] agree that SE added to sausages contributed to colour stability during storage. It is unlikely that microquantities of SE can modify sauce pH, despite SE being naturally acidic products, as seen for rosemary extracts [20]. The pH slightly increased in sauces without SE with storage. Nadeem et al. (2022) [67] also observed that pH slightly increases in chill-stored chicken nuggets with basil essential oil, concluding that this increase could be associated with lactic acid and protein degradation.

## 5. Conclusions

Sage distillation by-products are a valuable source of phenolic acids and flavonoids that can be revalorised as natural preservatives for food applications. In the present study, pasteurisation at 70 °C for 30 min and further chilling storage ensure the microbiological quality of yoghurt sauce and inhibition of microbial growth by SE cannot be evidenced, although SE70 shows some antimicrobial potential. Both SE have shown to have good antioxidant properties in the sauces. This finding is based on two results: (i) their main polyphenols, salvianic and rosmarinic acids, resist without degrading to mild pasteurisation conditions and are quite stable during shelf life; and (ii) SE improve radical scavenging capacity, delay primary and secondary lipid oxidation, and increase colour stability, contributing to sauce stabilisation. SE38 had a better antioxidant profile than SE70; therefore, the selection criteria for SE should be based on both the quantity and type of polyphenols. Due to their stability and antioxidant properties, sage polyphenols can be used to stabilise yoghurt sauces against oxidation. This fact also favours intake with sauce of dietary polyphenols with functional potential. Using SE makes it possible to control oxidative processes in yoghurt sauce emulsions, being an alternative for the formulation of clean label products.

## Figures and Tables

**Figure 1 antioxidants-12-00364-f001:**
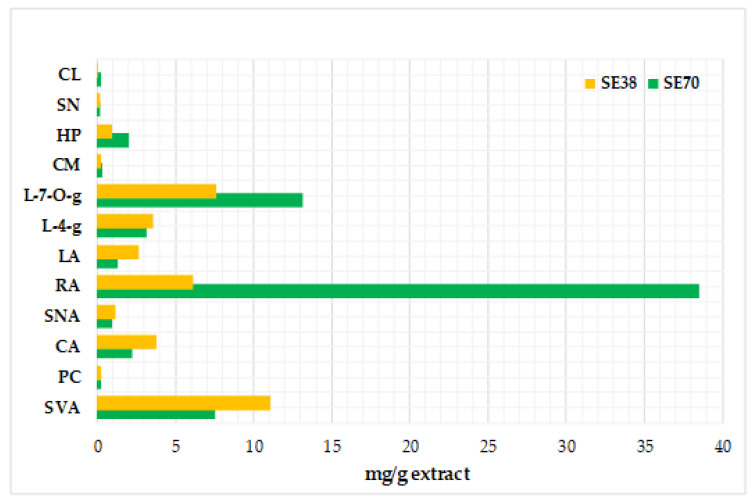
Polyphenol profile of sage extracts (SE38 and SE70) [14]. Abbreviations: SVA: salvianic acid; PC: protocatechuic acid; CA: caffeic acid; SNA: salvianolic acid; RA: rosmarinic acid; LA: lithospermic acid; L-4-g: luteolin-4-glucoside; L-O-g: luteolin-7-O-glucoronide; CM: cirsimaritin; HP: hesperidin; SN: salvigenin; CL: cirsileneol.

**Figure 2 antioxidants-12-00364-f002:**
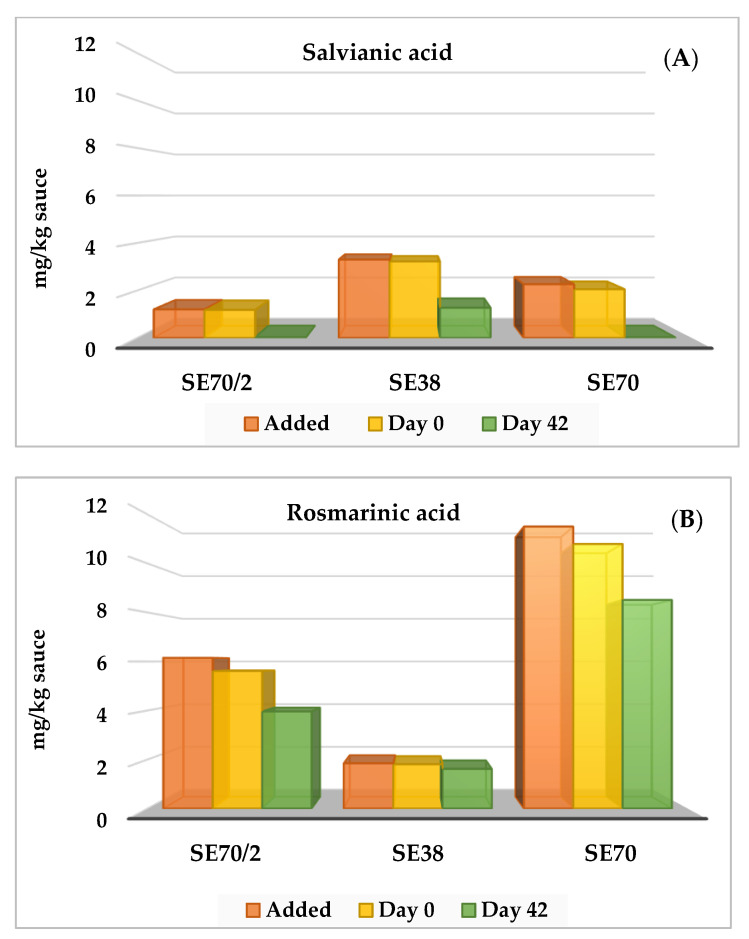
Added (raw sauce) and remaining quantities (mg/kg) of salvianic (**A**) and rosmarinic (**B**) acids in yoghurt sauces enriched with sage extracts (SE70/2, SE38 and SE70) and kept at 4 °C for 42 days. Standard Error of the Mean: Salvianic acid: 0.021 (SE70/2), 0.088 (SE38), and 0.018 (SE70); Rosmarinic acid: 0.204 (SE70/2), 0.034 (SE38), and 0.394 (SE70).

**Figure 3 antioxidants-12-00364-f003:**
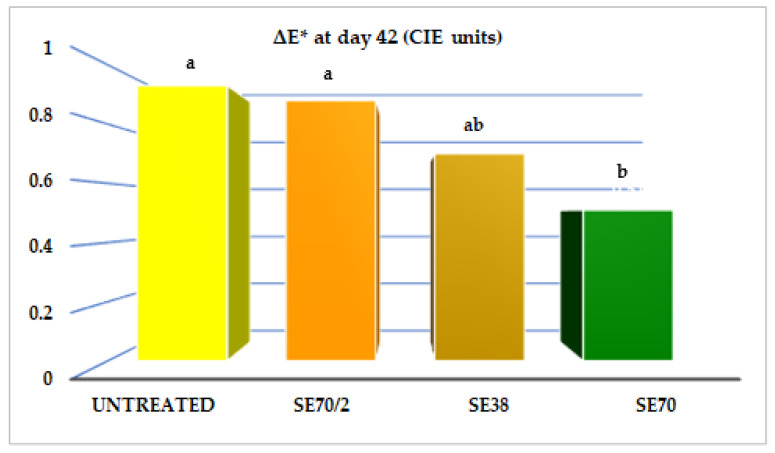
ΔE* colour values of yoghurt sauces enriched with sage extracts (SE70/2, SE38, and SE70) kept at 4 °C for 42 days. ΔE*= [(L*day0 − L*day42)^2^ + (a*day0 − a*day42)^2^ + (b*day0 − b*day42)^2^]^1/2^. Abbreviations: CIE: Commission Internationale de L’éclairage; L*: lightness; a*: Redness; b*: yellowness; ^a,b^ SE effects (One-way ANOVA; Tukey test, *p* < 0.05). Standard Error of the Mean: 0.08 (Untreated); 0.09 (SE70/2); 0.08 (SE38); and 0.06 (SE70).

**Table 1 antioxidants-12-00364-t001:** Yoghurt sauce ingredients.

Ingredients	g/100 g
Greek yoghurt	79.4
Honey	6.4
Inulin HSI (Beneo Orafti, Barcelona, Spain)	5.0
Sunflower oil	4.6
Dijon mustard	3.2
Water	1.0
Salt	0.3
**Additive**	**g/kg**
Sage extract *	≤0.3

* Added at 0.3 (SE38 and SE70) or 0.16 (SE70/2) g/kg raw sauce.

**Table 2 antioxidants-12-00364-t002:** Antioxidant capacity of sage extracts (SE70/2, SE70, and SE38) determined at the same concentrations used for 100 g of yoghurt sauce.

Sage Extracts	SE70/2	SE38	SE70		Comparisons ^(1)^	
	Mean	Mean	Mean	SEM	SE70/2	SE70
TPC (mg GAE/100 g)	12.48 ^b^	14.19 ^a^	13.94 ^a^	0.248	12%	2%
DPPH (mg TE/100 g)	4.59 ^c^	7.13 ^a^	6.23 ^a^	0.417	36%	13%
ABTS (mg TE/100 g)	7.32 ^c^	12.59 ^a^	8.65 ^b^	0.343	42%	31%
FRAP (µmol Fe ^2+^/100 g)	38.20 ^c^	78.30 ^a^	62.30 ^b^	3.820	51%	20%
**Tested concentrations**						
g extract/kg	0.16	0.30	0.30			
mg polyphenols/kg	11.29	11.29	20.95			

Abbreviations: SE: Sage extract; TPC: Total Phenolic Content; GAE: Gallic Acid Equivalent; ABTS: 2.2′-azinobis-(3-ethylbenzothiazoline-6-sulfonic); DPPH: 2,2-diphenyl-1-picrylhydrazyl radical; TE: Trolox Equivalents; FRAP: Ferric-reducing antioxidative power; SEM: Standard Error of the Mean. ^a,b,c^ SE effects (Tukey Test; *p* < 0.05). ^(1)^ Relative decreases respect to SE38 (100%).

**Table 3 antioxidants-12-00364-t003:** Microbial quality of yoghurt sauces enriched with sage extracts (SE70/2, SE38, and SE70) and kept at 4 °C for 42 days.

Yoghurt Sauces		Untreated	SE70/2	SE38	SE70	
	Day	Mean	Mean	Mean	Mean	SEM
Total viable counts (Log CFU/g)	0	<1	<1	<1	<1	
	42	1.78 ^a^	1.90 ^a^	1.72 ^a^	1.30 ^b^	0.173
Moulds and yeasts (Log CFU/g)	0	<1	<1	<1	<1	
	42	<1	<1	<1	<1	
*L. monocytogenes* (absence/25 g)	0	^n^.^d^.	^n^.^d^.	^n^.^d^.	^n^.^d^.	
	42	^n^.^d^.	^n^.^d^.	^n^.^d^.	^n^.^d^.	

Abbreviations: SE: sage extract; SEM: Standard Error of the Mean; ^n^.^d^.; not detected; ^a,b^ SE effects at the same storage time (One-way ANOVA; Tukey test; *p* < 0.05).

**Table 4 antioxidants-12-00364-t004:** Antioxidant capacity of yoghurt sauces enriched with sage extracts (SE70/2, SE38, and SE70) and kept at 4 °C for 42 days.

Yoghurt Sauces		Untreated	SE70/2	SE38	SE70		*p*-Values		
	Day	Mean	Mean	Mean	Mean	SEM	SE	S	SE × S
TPC (mg GAE/100 g)	0	38.82 ^c^	40.39 ^ab^	41.31 ^a^	40.98 ^a^	0.301	***	^N^.^S^.	^N^.^S^.
	42	38.61 ^c^	40.14 ^b^	41.04 ^a^	41.24 ^a^				
DPPH (mg TE/100 g)	0	7.44 ^c^	10.90 ^ab^	11.54 ^ab^	12.20 ^a^	0.475	***	**	***
	42	4.98 ^d^	10.36 ^b^	11.94 ^a^	11.63 ^ab^				
ABTS (mg TE/100 g)	0	7.72 ^d^	11.89 ^b^	14.59 ^a^	12.63 ^b^	0.229	***	***	***
	42	5.54 ^e^	9.18 ^c^	9.56 ^c^	9.55 ^c^				
FRAP (µmol Fe^2+^/100 g)	0	86.57 ^d^	114.87 ^c^	128.23 ^b^	139.74 ^a^	2.448	***	***	***
	42	64.72 ^e^	94.38 ^d^	114.45 ^c^	107.88 ^c^				

Abbreviations: SE: sage extract; S: storage; P: probability; SEM: Standard Error of the Mean; TPC: Total Phenolic Content; GAE: Gallic Acid Equivalent; ABTS: 2.2′-azinobis-(3-ethylbenzothiazoline-6-sulfonic); DPPH: 2,2-diphenyl-1-picrylhydrazyl radical; TE: Trolox Equivalents; FRAP: Ferric-reducing antioxidative power. ^a–e^ Effects of SE addition and storage time (Two-way ANOVA; Tukey test, *p* < 0.05). SE × S: Interactions; Significance levels: *** *p* < 0.001; ** *p* < 0.01; *p* < 0.05; ^N^.^S^. *p* > 0.05.

**Table 5 antioxidants-12-00364-t005:** Lipid oxidation indexes, CIELab colour, and pH values of yoghurt sauces enriched with sage extracts (SE70/2, SE38, and SE70) and kept at 4 °C for 42 days.

Yoghurt Sauces		Untreated	SE70/2	SE38	SE70		*p*-Values		
	Day	Mean	Mean	Mean	Mean	SEM	SE	S	SE × S
CD (µmol/g)	0	1.75 ^a^	0.93 ^de^	1.46 ^b^	1.00 ^cd^	0.069	***	***	***
	42	1.21 ^c^	0.89 ^de^	0.78 ^e^	0.92 ^de^				
CT (µmol/g)	0	0.77 ^a^	0.29 ^cd^	0.68 ^b^	0.31 ^c^	0.016	***	***	***
	42	0.25 ^cde^	0.23 ^de^	0.24 ^cde^	0.21 ^e^				
CD + CT (µmol/g)	0	2.51 ^a^	1.21 ^cde^	2.13 ^b^	1.31 ^cd^	0.080	***	***	***
	42	1.46 ^c^	1.10 ^de^	1.02 ^e^	1.13 ^de^				
PV (meq O_2_/kg)	0	1.85 ^c^	1.14 ^c^	1.29 ^c^	1.19 ^c^	0.346	***	***	***
	42	14.08 ^a^	7.34 ^b^	7.52 ^b^	7.58 ^b^				
TBARS (mg MDA/kg)	0	0.44 ^c^	0.46 ^c^	0.45 ^c^	0.45 ^c^	0.016	***	***	***
	42	0.67 ^a^	0.53 ^b^	0.55 ^b^	0.55 ^b^				
L* (CIE units)	0	87.06 ^a^	86.44 ^bc^	86.11 ^cd^	86.06 ^cd^	0.156	***	***	^N^.^S^.
	42	86.78 ^ab^	85.94 ^d^	85.70 ^d^	85.70 ^d^				
a* (CIE units)	0	−1.66 ^e^	−1.42 ^d^	−1.14 ^bc^	−1.15 ^bc^	0.071	***	***	^N^.^S^.
	42	−1.30 ^cd^	−0.97 ^ab^	−0.87 ^a^	−0.97 ^ab^				
b* (CIE units)	0	11.50 ^c^	11.92 ^ab^	12.10 ^a^	12.01 ^a^	0.072	***	***	***
	42	10.78 ^d^	11.43 ^c^	11.84 ^b^	11.93 ^ab^				
pH	0	3.92 ^c^	4.03 ^abc^	4.01 ^abc^	3.94 ^bc^	0.022	^N^.^S^.	***	***
	42	4.08 ^abc^	4.11 ^a^	4.09 ^ab^	4.11 ^a^				

Abbreviations: SE: sage extract; S: storage; P: probability; SEM: Standard Error of the Mean; CD: Conjugated dienes; CT: Conjugated trienes; PV Peroxide value; TBARS: Thiobarbituric acid reactive substances; MDA: malondialdehyde; CIE: Commission Internationale de L’éclairage; L*: lightness; a*: Redness; b*: yellowness. ^a–e^ Effects of SE addition and storage time (Two-way ANOVA; Tukey test, *p* < 0.05). SE × S: Interactions; Significance levels: *** *p* < 0.001; ^N^.^S^. *p* > 0.05.

## Data Availability

Data are contained within the article.

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
