# Peer review of "Polyphenol Extracts from Sage (Salvia lavandulifolia Vahl) By-Products as Natural Antioxidants for Pasteurised Chilled Yoghurt Sauce"

_antioxidants, 2023, doi:10.3390/antiox12020364_

Round 1

Reviewer 1 Report

The manuscript presents an interesting study on the use of sage extracts as an additive to yogurt sauces. In my opinion, this topic fits the scientific profile of the Antioxidants journal, although the text needs to be revised and missing information filled in before the manuscript is accepted for publication. My suggestions are as follows:

·         Please indicate throughout the manuscript whether sage by-product extracts (e.g. title and line 16) or sage extracts (e.g. line 117) were obtained and studied.

·         Please justify the advisability of testing the addition of sage extracts to low-fat yoghurt sauces (line106).

·         Please briefly describe how sage extracts are obtained (line 122).

·         Have you used methanolic extracts for food (e.g. lines 123, 164)? Is it safe? Is it legal?

·         Please explain the discrepancies between the declared composition of the yoghurt sauce (Table 1) and its approximate composition (lines 152-156). What is the reason for such large differences in the content of individual ingredients, for example fat? Please explain why the composition shown is “the approximate composition” when the analyzes were performed according to ISO standards and AOAC methods (line 155). Please comment.

·         Did you use external standards for rosmarinic and salvianic acids acid during HPLC-DAD analysis (lines 164-171)? Please comment.

·         Please comment in detail on the differences in antioxidant capacity of sage extracts (SE70/2, SE70, and SE38) shown in Table 2. What was the reason for these differences? Were these differences as expected? Please comment and discuss.

·         Please correct "L. Monocytogenes" to "L. monocytogenes" (line 234).

·         What was the reason for the observed differences in total plate counts on day 42 of the study (Table 3)? Were these differences as expected? Please comment and discuss.

·         Please provide the standard deviation value in addition to the mean values ​​in Figure 2. Please attach the results of the statistical analysis of these results. What was the reason for the observed differences? Were these differences as expected? Please comment and discuss.

·         In my opinion, a two-way ANOVA (additive and time) would be useful to analyze the results presented in Table 4. Please comment.

·         In my opinion, a two-way ANOVA (additive and time) would be useful to analyze the results presented in Table 5. Please comment. What were the color values ​​of the sage extracts used? Please comment and discuss. Why was no statistical evaluation carried out for the pH values? The initial pH of the samples (Table 5) was similar to the untreated samples, is this possible? What was the pH of used sage extracts? Mean values ​​and standard deviations of these pH values ​​should be presented in the text. What was the reason for the observed differences in these values ​​of the stored samples? Were these differences as expected? Please comment and discuss.

·         In Figure 3, please indicate the value of the standard deviation in addition to the mean values. What was the reason for the observed differences in the stored sample values? Were these differences as expected? Please comment and discuss.

·         Please discuss the use of sage extracts as an additive to yoghurt sauces for higher fat content. Was it relevant that the yogurt sauces you tested were low-fat? Please comment and discuss.

Reviewer 2 Report

The manuscript reports an interesting research. Some aspects could be improve. Specific observation are reported below.

Introduction

Lines 74-76. The statement could be slightly change. The studies reported by authors referred about antimicrobial activity. Although antioxidant activity generally exerts an antimicrobial effect, it would be more correct to emphasize the activity of SE also as an antimicrobial.

Lines 77-81. This part could be summarize or omitted. The study’s topic is  yoghurt sauce, the classification of the different sauces is not important.

Material and methods

2.1. Experimental design

Lines 104-106. Authors should report the origin of the two SE (SE38 and SE70). Were they obtained from two different varieties? Other?

2.2. Obtention and typification of sage extracts.

Even if the reference reports the procedure for obtaining the extract, in order to make the work more readable, it would be appropriate for the authors to define (at least briefly) the type of SE: distillate, extract with solvent or other.

Line 125. The identification mode could be describe better, at least from a lexical point of view. Spectra recorded at different retention times were compared to spectra of pure compounds, I suppose.

Figure 1. Legend. The last phrase: “Phenolic acids / flavonoids ratio: SE38: 25.07/12.57 (2.1); SE70: 50.79/19.03 (3.0).” is a duplication of what is already reported in the text. It could be omitted.

2.3. Elaboration of yoghurt sauce

Authors should describe the pasteurisation process. It is important know the dimension of bottles/jars, how they were heated, if the temperature of 70°C refers to the bath or at the bottle/jar (centre), the same is for time, the stirring is for water bath. Other particulars that could describe the heating process should be reported. The best solution would be report the heat penetration curve to the cold spot. Finally why 70 °C for 30 minutes? Is it a rule? Many of this information are not reported in reference [21].

Table 1. Check the unit of measure. It is very strange the ingredients were reported as g/kg. The sum does not make one kg.

Line 152. It is not clear if at initial time and after 42 days at 4° samples were put at – 80°C.

2.5. Antioxidant capacity and polyphenols analyses

Lines 170-171. It is not clear which tests the authors are referring to. Reference [14] reports some that are not reported in the present manuscript.

Results

3.3. Stability of salvianic and rosmarinic acids and figure 2.

Mainly in figure 2, authors should explain that row sauces are as non-pasteurized or before pasteurization.

3.4. Antioxidant capacity

Line 274. The statement is not clear. What authors meant as “profile” and what are referred the “11 mg polyphenols/kg”). Could authors explain?

3.5. Oxidative stability

Lines 310-311. The last phrase could be omitted or transpose after colour result description.

Discussion

Lines 318-322. The two phrases are repetitions. I suggest to omit the first one with the reference [41].

Lines 339-340. The statement is not clear. The previous study reported three concentration of SE38 and SE70. Authors could better explain what they would mean.

Lines 429-431. From text, authors stated MDA was not detected after thermal treatment. From the table 5, although statistically significant, MDA is detected at time zero.

Lines 348, 469-470 and other parts of the manuscript. After clarified the pasteurisation process (as requested previously: comment at 2.3. Elaboration of yoghurt sauce), the thermal process parameters reported in these points of the discussion could be modify or explain.

Round 2

Reviewer 1 Report

The authors have corrected the original version of the manuscript to my satisfaction. However, a small error crept in during proofreading, which must be corrected before the current version of the manuscript is accepted for publication:

·         Species names of microorganisms must be written in lowercase: „P. areuginosa” and „B. cereus” (see line 417).

Author Response

This mistake has been amended. Thank you for your review.

Reviewer 2 Report

Authors have modified text as the observations/comments required. In this form the manuscript can be published.

Author Response

Thank you for your review.